

# Universal geometry of two-neutron halos and Borromean Efimov states close to dissociation

Pascal Naidon⋆

Few-Body Systems Physics Laboratory, RIKEN Nishina Centre, RIKEN, Wakō, 351-0198 Japan

⋆ pascal@riken.jp

## Abstract

The geometry of Borromean three-body halos, such as two-neutron halo nuclei or triatomic molecules close to dissociation, is investigated using a three-body model. This model enables to analytically derive the universal geometric properties found recently within an effective-field theory for halos made of a core and two resonantly-interacting particles [Phys. Rev. Lett., 128, 212501 (2022)]. It is shown that these properties not only apply to the ground three-body state, but also to all the excited (Efimov) states where the core-particle interaction is resonant. Furthermore, a universal geometry persists away from the resonant regime between the two particles, for any state close to the three-body threshold. This "halo universality", which applies equally to all states, is different from the Efimov universality, which is only approximate for the ground state. It is explained by the separability of the hyper-radius and hyper-angles close to the three-body dissociation threshold.



## 1   Introduction

Quantum halos [1], i.e. quantum few-body bound states whose spatial extent exceeds the range of the bodies' interactions, have been studied for over four decades since the experimental discovery of halos in atomic nuclei in the 1980s [2–5], followed by the controlled creation of halos in ultracold-atom experiments from the 2000s [6–9]. Quantum halo systems can be composed of identical particles loosely bound to each other, or as is often the case for halo nuclei, a composite core and a few loosely bound particles forming the halo. Moroever, quantum halos can be Borromean [10], i.e. they do not remain bound if one of the particles is removed. The geometry of quantum halos is characterised by large mean square radii [11–17], which can be extracted from experimental measurements [18–23]. In a recent work based on an effective-field theory [24], universal analytical relations were found between different mean square radii for Borromean three-body halos made of a core and two resonantly-interacting particles, such as two-neutron halo nuclei.

In the present work, it is shown how the analytic relations can be obtained from the Faddeev approach to the three-body problem. Although it is implicit in the work of Ref. [24], it is here emphasised that the analytical relations apply not only to a ground state but also to excited Borromean halo states. The applicability of the relations is then tested numerically within a separable three-body model where the three-body Efimov effect occurs. These numerical calculations confirm that the analytic properties are relevant to any Borromean halo state, i.e. any Efimov state close to the three-body dissociation. Finally, the limiting values close and away from the two-particle resonance are retrieved analytically in the hyper-spherical representation.

## 2   Model

We start with a three-body model for a core particle denoted by 3, of mass $m_3$, interacting with two identical particles denoted by 1 and 2, of mass $m_1 = m_2$ – see Fig. 1. The corresponding Schrödinger equation for the three-body wave function $\tilde{\Psi}$ in momentum representation reads

$$\left( \sum_{i=1}^{3} \frac{\hbar^2 k_i^2}{2m_i} + \sum_{i<j} \hat{V}_{ij} - E \right) \tilde{\Psi} = 0, \tag{1}$$

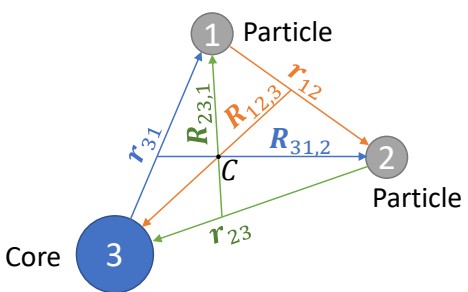

Figure 1: Schematic representation of a system formed by a core (3) and two identical particles (1 and 2).

where $E$ is the total energy and $\hat{V}_{ij}$ are the respective two-body interaction operators, having ranges denoted as $\Lambda_{ij}^{-1}$. In the present notation, an operator $\hat{O}_{ij}$ acts on the relative wave vector $\boldsymbol{k}_{ij} = \frac{m_i \boldsymbol{k}_j - m_j \boldsymbol{k}_i}{m_i + m_j}$ between particles $i$ and $j$. Introducing the Faddeev components [25] $\mathcal{F}_{ij} = \hat{V}_{ij}\tilde{\Psi}$, one can write for $E < 0$,

$$\tilde{\Psi} = \frac{\mathcal{F}_{12} + \mathcal{F}_{23} + \mathcal{F}_{31}}{E - \sum_{i=1}^{3} \frac{\hbar^2 k_i^2}{2m_i}} . \tag{2}$$

It follows from the definition of the Faddeev components that they satisfy the following Faddeev equations:

$$\mathcal{F}_{ij} = \hat{T}_{ij}\left(z_{ij}\right) \frac{\mathcal{F}_{jk} + \mathcal{F}_{ki}}{E - \sum_{i=1}^{3} \frac{\hbar^2 k_i^2}{2m_i}} , \tag{3}$$

where the two-body transition operators $\hat{T}_{ij}(z)$ are defined from the original interactions $\hat{V}_{ij}$ by:

$$\hat{T}_{ij}(z) = \hat{V}_{ij} + \left(\hat{V}_{ij} G_{ij}^+(z)\right) \hat{T}_{ij}(z), \tag{4}$$

where $G_{ij}^+(z) = \left(z + i0^+ - \frac{\hbar^2}{2\mu_{ij}} k_{ij}^2\right)^{-1}$ and $\mu_{ij} = \left(\frac{1}{m_i} + \frac{1}{m_j}\right)^{-1}$ is the reduced mass for particles $i$ and $j$. The two-body energy $z_{ij}$ in Eq. (3) is obtained by subtracting from the total energy the centre-of-mass kinetic energy and the relative kinetic energy between the dimer $(i,j)$ and particle $k$,

$$z_{ij} = E - \frac{\hbar^2}{2(m_1 + m_2 + m_3)} K_C^2 - \frac{\hbar^2}{2\mu_{ij,k}} K_{ij,k}^2 , \tag{5}$$

where $\boldsymbol{K}_C = \boldsymbol{k}_1 + \boldsymbol{k}_2 + \boldsymbol{k}_3$ is the total wave vector, $\mu_{ij,k} = \left(\frac{1}{m_i + m_j} + \frac{1}{m_k}\right)^{-1}$ is the reduced mass and $\boldsymbol{K}_{ij,k} = \frac{(m_i + m_j)\boldsymbol{k}_k - m_k(\boldsymbol{k}_i + \boldsymbol{k}_j)}{m_i + m_j + m_k}$ is the relative wave vector between the dimer $(i,j)$ and particle $k$. Since the system is translationally invariant, it will be assumed in the following that the total wave vector $\boldsymbol{K}_C$ is zero.

## 3 Mean square radii

Following Ref. [24], let us denote the mass ratio $m_3/m_1$ as $A$, and define the following mean square *matter radius* $\langle r_m^2 \rangle$ and mean square *core radius* $\langle r_c^2 \rangle$ (or *charge radius,* if the core carries

an electric charge)

$$\langle r_m^2 \rangle \equiv \frac{2\langle r_{C1}^2 \rangle + A\langle r_{C3}^2 \rangle}{A+2}, \tag{6}$$

$$\langle r_c^2 \rangle \equiv \langle r_{C3}^2 \rangle, \tag{7}$$

where $C$ denotes the centre of mass of the three particles, and $\boldsymbol{r}_{Ci} = \boldsymbol{r}_i - \boldsymbol{r}_C$ is the relative vector between particle $i$ and the centre of mass $C$. One can easily show that

$$r_{C1} = \frac{A+1}{A+2} R_{23,1}, \tag{8}$$

$$r_{C3} = \frac{2}{A+2} R_{12,3}, \tag{9}$$

where $R_{ij,k}$ is the relative vector between particle $k$ and the centre of mass of the dimer $(i,j)$ - see Fig. 1. It follows that

$$\frac{\langle r_m^2 \rangle}{\langle r_c^2 \rangle} = \frac{\frac{1}{2}(A+1)^2 \frac{\langle R_{23,1}^2 \rangle}{\langle R_{12,3}^2 \rangle} + A}{A+2}. \tag{10}$$

The two averages $\langle R_{23,1}^2 \rangle$ and $\langle R_{12,3}^2 \rangle$ may be calculated from the wave function $\tilde{\Psi}$ in momentum representation as

$$\langle R_{23,1}^2 \rangle = \frac{\int d^3 K_{23,1} d^3 k_{23} \left| \boldsymbol{\nabla}_{K_{23,1}} \tilde{\Psi} \right|^2}{\int d^3 K_{23,1} d^3 k_{23} \left| \tilde{\Psi} \right|^2}, \tag{11}$$

$$\langle R_{12,3}^2 \rangle = \frac{\int d^3 K_{12,3} d^3 k_{12} \left| \boldsymbol{\nabla}_{K_{12,3}} \tilde{\Psi} \right|^2}{\int d^3 K_{12,3} d^3 k_{12} \left| \tilde{\Psi} \right|^2}. \tag{12}$$

## 4 Two-particle resonance

Let us now consider a bound core-particle-particle system (in a ground or excited state) close to the three-body dissociation threshold $E \to 0^-$, so that it becomes a halo whose extent exceeds the ranges $\Lambda_{ij}^{-1}$ of the particles' interactions. In this situation, the calculation of the square radii is dominated by the low-momentum part of the wave function, $k, K \ll \Lambda_{ij}$. At low momenta and energy, the (on- and off-shell) two-body T-matrix elements are given by[1]

$$\langle \boldsymbol{k} | \hat{T}_{ij}(z_{ij}) | \boldsymbol{k}' \rangle \approx \frac{4\pi\hbar^2}{2\mu_{ij}} \left( \frac{1}{a_{ij}} + i \sqrt{\frac{2\mu_{ij}}{\hbar^2} z_{ij}} \right)^{-1}, \tag{13}$$

where $a_{ij}$ is the s-wave scattering length between particles $i$ and $j$. It follows from Eq. (3) that each Faddeev component $\mathcal{F}_{ij}$ at low momenta is proportional to the right-hand side of Eq. (13).

Close to a resonance between particles 1 and 2, such that $|a_{12}^{-1}| \ll |a_{23}^{-1}|, |a_{31}^{-1}|, \Lambda_{ij}$, and for sufficiently small energy $\sqrt{2\mu_{12}|E|}/\hbar \ll |a_{23}^{-1}|, |a_{31}^{-1}|$ of the three-body system, the Faddeev component $\mathcal{F}_{12}$ at low momenta $k_{12} \ll \Lambda_{12}$ and $K_{12,3} \lesssim \sqrt{2\mu_{12,3}|E|}/\hbar \ll \Lambda_{12}$ becomes

---

[1]See Appendix A for details.

dominant over $\mathcal{F}_{23}$ and $\mathcal{F}_{31}$ in the three-body wave function of Eq. (2). Furthermore, $\mathcal{F}_{12}$ is proportional to $\langle 0|\hat{T}_{12}|0\rangle$ and depends only on $K_{12,3}$:

$$\mathcal{F}_{12} \propto \mathcal{F}(K_{12,3}) = \frac{4\pi\hbar^2/2\mu_{12}}{\frac{1}{a_{12}} - \sqrt{-\frac{2\mu_{12}}{\hbar^2}E + \frac{\mu_{12}}{\mu_{12,3}}K_{12,3}^2}} \, . \tag{14}$$

The calculation of the square radii is thus simplified and yields

$$\frac{\langle r_m^2 \rangle}{\langle r_c^2 \rangle} = \frac{A}{2}\left(1 + \frac{f_n}{f_c}\right), \tag{15}$$

where the quantities $f_n$ and $f_c$ are determined solely by the Faddeev component $\mathcal{F}$ associated with the two resonant particles,

$$f_n \equiv \frac{1}{2}\int d^3K \frac{\mathcal{F}(K)^2}{\tilde{K}^3}, \tag{16}$$

$$f_c \equiv \int d^3K \left(\frac{\mathcal{F}'(K)^2}{\tilde{K}} - \frac{K\mathcal{F}(K)\mathcal{F}'(K)}{\tilde{K}^3} + \frac{K^2\mathcal{F}(K)^2}{2\tilde{K}^5}\right), \tag{17}$$

with $\tilde{K}^2 = K^2 - \frac{2\mu_{12,3}E}{\hbar^2}$. For $a_{12} < 0$ and $E < 0$, it can be shown[2] that $f_n$ and $f_c$ reduce to the following integrals

$$f_n(\beta) \propto \int_1^\infty dy \frac{\sqrt{y-1}}{2y^{3/2}\left(\beta + \sqrt{y}\right)^2}, \tag{18}$$

$$f_c(\beta) \propto \int_1^\infty \frac{dy}{2\sqrt{y(y-1)}\left(\beta + \sqrt{y}\right)^2}, \tag{19}$$

where

$$\beta = \sqrt{\frac{E_{12}}{|E|}}, \tag{20}$$

is the square root ratio of the particle-particle virtual energy $E_{12} = \frac{\hbar^2}{2\mu_{12}|a_{12}|^2}$ and the trimer binding energy $|E|$. The above integrals were shown in Ref. [24] to admit the following analytical expressions,

$$f_n(\beta) \propto \begin{cases} \beta^{-3}\left(\pi - 2\beta + (\beta^2 - 2)\frac{\arccos\beta}{\sqrt{1-\beta^2}}\right), & \beta < 1, \\[4mm] \beta^{-3}\left(\pi - 2\beta + (\beta^2 - 2)\frac{\text{arccosh}\beta}{\sqrt{\beta^2-1}}\right), & \beta > 1, \end{cases} \tag{21}$$

$$f_c(\beta) \propto \begin{cases} \dfrac{1}{1-\beta^2} - \dfrac{\beta\arccos\beta}{(1-\beta^2)^{3/2}}, & \beta < 1, \\[4mm] -\dfrac{1}{\beta^2-1} + \dfrac{\beta\,\text{arccosh}\beta}{(\beta^2-1)^{3/2}}, & \beta > 1. \end{cases} \tag{22}$$

Thus, the ratio of matter and charge radii of Eq. (15) is universally determined by the mass ratio $A$ and the square root ratio $\beta$. In particular, it tends to $A$ for $\beta \gg 1$ and to $\frac{2}{3}A$ for $\beta \ll 1$.

---

[2]See Appendix B for details.

More generally, any geometric property of the system that does not depend explicitly on its size, such as length ratios and angles, is universally determined by $A$ and $\beta$, owing to the form of Eq. (14) and the fact that $\mathcal{F}_{23}, \mathcal{F}_{31} \ll \mathcal{F}_{12}$. The word *universal* here means independent of the details of the interactions, but it also means independent on whether the considered state is excited or not, as we shall confirm numerically below.

# 5 Numerical investigation

In order to test the validity of the previous analytical results, the system is now investigated numerically close to and far from the two-body resonance, for states of total angular momentum equal to zero. The two particles are assumed to be in a symmetric state, i.e. they correspond to either two identical bosons, or two identical fermions in antisymmetric spin states, such as two neutrons in a singlet state. For simplicity, the interactions are taken to be separable, $\langle \boldsymbol{k} | \hat{V}_{ij} | \boldsymbol{k}' \rangle = g_{ij} \phi_{ij}(\boldsymbol{k}) \phi_{ij}(\boldsymbol{k}')$. In this case, the transition operators $\hat{T}_{ij}$ are also separable, and the Faddeev Eqs. (3) simplify to integral equations of the Skorniakov−Ter-Martirosian (STM) type [26], which can easily be solved numerically. The form factors $\phi_{ij}$ are chosen to be of the Gaussian type, $\phi_{ij}(\boldsymbol{k}) = \exp(-k^2/\Lambda_{ij}^2)$, where $\Lambda_{ij}^{-1}$ characterise the range of interactions. The interactions being of the same physical nature, $\Lambda_{ij}$ are taken for simplicity to be all equal to the same order of magnitude $\Lambda$. The strengths $g_{12}$ and $g_{23} = g_{31}$ determine the respective scattering lengths $a_{12}$ and $a_{23} = a_{31}$.

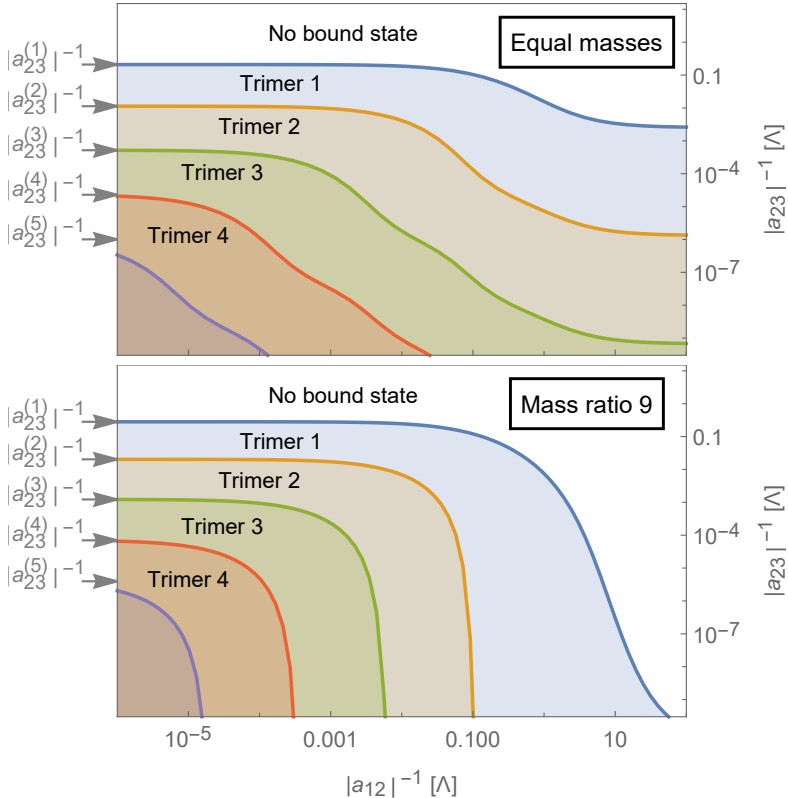

Figure 2: Regions of existence of Borromean bound states of a core and two identical particles, as a function of the core-particle inverse scattering length $|a_{23}|^{-1}$ and particle-particle inverse scattering length $|a_{12}|^{-1}$. Top panel: equal mass case $A = 1$. Bottom panel: heavy core and light particle, with mass ratio $A = 9$. The arrows show the critical core-particle scattering lengths $a_{23}^{(n)}$ to get an $n$th bound state.

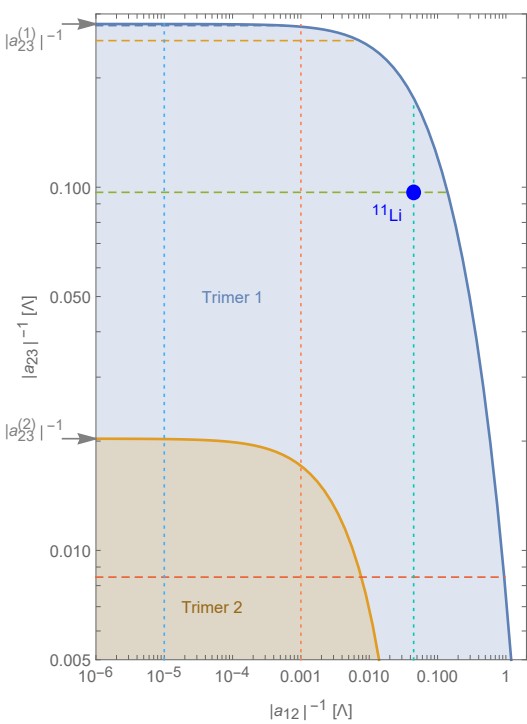

Figure 3: Close-up figure of the bottom panel of Fig. 2. The horizontal dashed lines correspond to the values $|a_{23}|^{-1} = 0.99, 0.9, 0.344, 0.03$ in units of $|a_{23}^{(1)}|^{-1}$. The vertical dotted lines correspond to the values $|a_{23}|^{-1} = 10^{-5}, 10^{-3}, 0.0447$ in units of $\Lambda$.

Figure 2 shows the regions of existence of three-body bound states as a function of $1/a_{12} < 0$ and $1/a_{23} < 0$, i.e. in the Borromean region where there are no two-body bound states. One can see that the core-particle scattering length $|a_{23}|$ must exceed a critical value $|a_{23}^{(1)}|$ in order to allow a first three-body bound state. Owing to the Efimov effect [15, 27–30], there is an infinite number of three-body bound states as $|a_{23}|$ is further increased towards infinity. The curves in Fig. 2 represent the sucessive thresholds for the appearance of these trimer states. Conversely, these curves can also be regarded as the thresholds where the trimer states dissociate into three unbound particles when the interactions are weakened. It is near these thresholds that the trimer becomes a large halo. Due to the Efimov effect, the thresholds follow a geometric progression in the limit of highly-excited states. However, for large mass ratio $A$ and small scattering length $|a_{12}|$, the discrete scaling factor is so large that only the ground state is observable, while the excited states are too weakly bound to be seen. We thus focus on the region of large scattering length $|a_{12}|$ relevant to two identical particles close to unitarity such as two neutrons.

This region is shown for $A = 9$ in Fig. 3, where the ground-state and first-excited state are visible. Let us first scan the ground state by varying the particle-particle scattering length $a_{12}$ as indicated by the horizontal dashed lines, and calculate its geometric properties such as the matter/core radius ratio $\langle r_m^2 \rangle / \langle r_c^2 \rangle$ using Eqs. (11-12). The result is shown in the top panel of Fig. 4. The analytical formula based on Eqs. (15, 21, 22) found in Ref. [24] accurately reproduces the numerical calculations whenever the system is close enough to the dissociation threshold. In particular, for large particle-particle scattering length $|a_{12}|$, the ratio approaches the limit $\frac{2}{3}A$ when $|a_{23}|$ is close (within 1%) to the critical value $|a_{23}^{(1)}|$, whereas for smaller values of $|a_{12}|$ the ratio approaches the limit $A$ near the threshold, regardless of the core-particle scattering length $a_{23}$.

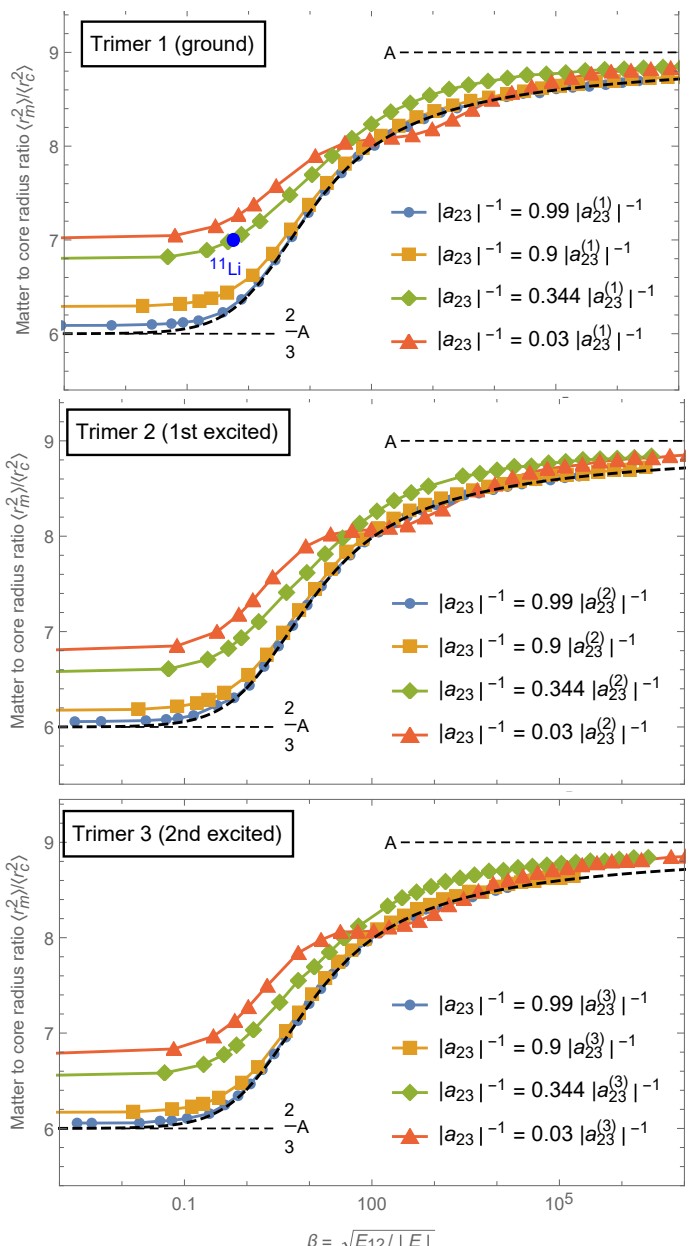

Figure 4: Matter/Core radius ratio $\langle r_m^2 \rangle / \langle r_c^2 \rangle$ as a function of the square root ratio $\beta$ of the two-particle virtual binding energy $E_{12}$ and the trimer binding energy $|E|$. The different curves correspond to different values of $a_{23}$, which for the ground-state trimer are shown by horizontal dashed lines in Fig. 3. The dashed curve corresponds to the analytical formula given by Eqs. (15, 21, 22) and the horizontal dotted lines show the limits $\frac{2}{3}A$ and $A$. Top panel: ground-state trimer ; middle panel: first excited trimer. Bottom panel: second excited trimer.

The same situation is observed for excited states, as shown in the middle and bottom panels of Fig. 4. The results for the first two excited states look nearly identical, which is expected since they follow a discrete scaling invariance associated with the Efimov effect, whereas small differences can be seen for the ground state, which is also expected since it deviates more strongly from the discrete scaling invariance. However, the differences remain small, and the results are identical for all states close to the dissociation threshold $E \to 0^-$, i.e. either $|a_{23}| \approx |a_{23}^{(i)}|$ or $\beta \gg 1$.

# 6 Hyper-spherical representation

The universality of the geometry close to the dissociation threshold can be understood using the hyper-spherical coordinates constituted by the hyper-radius $R = \sqrt{x_k^2 + y_k^2}$ giving the global size of the trimer and the hyper-angles, such as $\alpha_k = \arctan \frac{y_k}{x_k}$, describing its shape, where $\boldsymbol{x}_k = \left(\mu_{ij}/m\right)^{1/2} \boldsymbol{r}_{ij}$, $\boldsymbol{y}_k = \left(\mu_{ij,k}/m\right)^{1/2} \boldsymbol{R}_{ij,k}$, and $m$ is a normalisation mass which can be taken to be the particles' mass $m_1 = m_2$. In these coordinates, the wave function $\Psi$ of a halo state with zero total angular momentum admits the following hyper-spherical adiabatic expansion [31],

$$\Psi = \sum_{n=1}^{\infty} \frac{F_n(R)}{R^2} \left[ \sum_{i=1}^{3} \Phi_n^{(i)}(\alpha_i; R) \right], \tag{23}$$

where at large distances $R \gg \Lambda^{-1}$ the hyper-radial functions $F_n(R)$ are solutions of

$$\left( -\partial_R^2 + \frac{s_n^2(R) - 1/4}{R^2} - \frac{2mE}{\hbar^2} \right) \sqrt{R} F_n(R) = 0, \tag{24}$$

and the hyper-angular wave functions $\Phi_n^{(i)}$ are given by [29, 30]

$$\Phi_n^{(i)}(\alpha; R) = \lambda^{(i)} \frac{\sin\left[ s_n(R)\left(\frac{\pi}{2} - \alpha\right)\right]}{\sin 2\alpha}, \tag{25}$$

where $s_n(R)$ are the solutions of

$$\left( -\cos\left(s\frac{\pi}{2}\right) + \frac{2}{s}\frac{\sin(s\gamma)}{\sin 2\gamma} + \frac{\sin\left(s\frac{\pi}{2}\right)}{s}\frac{R}{a_{23}} \right)\left( -\cos\left(s\frac{\pi}{2}\right) + \frac{\sin\left(s\frac{\pi}{2}\right)}{s}\frac{R}{a_{12}} \right) = 2\left( \frac{2\sin(s\gamma')}{s\sin 2\gamma'} \right)^2, \tag{26}$$

with $\gamma = \arcsin\frac{1}{1+A}$ and $\gamma' = \frac{\pi}{4} - \frac{\gamma}{2}$.

For $1/|a_{12}| \neq 0$, the solutions $s_n(R) \xrightarrow{R\to\infty} 2n$. In this case, the trimer's extent diverges logarithmically with vanishing binding energy in the channel $n = 1$, while it remains finite in other channels, as discussed long ago in Ref. [11]. As a result, the wave function is dominated by the channel $n = 1$ at large $R$, and from $s_1(R) \to 2$ one can check from Eq. (25) that it does not depend on the hyper-angles. It follows that $\langle y_1^2 \rangle = \langle y_3^2 \rangle$ [11], thus $\langle R_{23,1}^2 \rangle / \langle R_{12,3}^2 \rangle = \left(\mu_{23,1}/\mu_{12,3}\right)^{1/2}$, and from Eq. (10) one recovers the limit $\langle r_m^2 \rangle / \langle r_c^2 \rangle = A$ seen in the right part of Fig. 4. The logarithmic divergence of the mean square radii in this limit is illustrated by the three lowest curves (circles, squares, and diamonds) in Fig. 5 corresponding to finite values of $a_{12}$.

The same hyper-spherical analysis can be carried out at the two particles' resonance $1/|a_{12}| = 0$. In that case, the solutions $s_n(R) \xrightarrow{R\to\infty} n$. As a result, the channel $n = 1$ is again dominant at large hyper-radius, but it now leads to a divergence of the trimer's extent that is slightly slower than the inverse of the energy. This divergence is illustrated by the top curve (triangles) in Fig. 5 corresponding to an infinite $a_{12}$. In this limit, the hyper-radius and hyper-angles separate again, but there is now a dependence on the hyper-angles. From the hyper-angular functions $\Phi_1^{(i)}$, one can recover[3] the ratio $\langle r_m^2 \rangle / \langle r_c^2 \rangle = \frac{2}{3}A$ seen in the left part of Fig. 4.

It is important to note that the above results rely only on the behaviour of $s_n^2(R)$ at large hyper-radius $R \gg |a_{12}|, |a_{23}|$, where the hyper-radial potential $[s_n^2(R) - 1/4]/R^2$ of Eq. (24)

---

[3]See Appendix C for details.

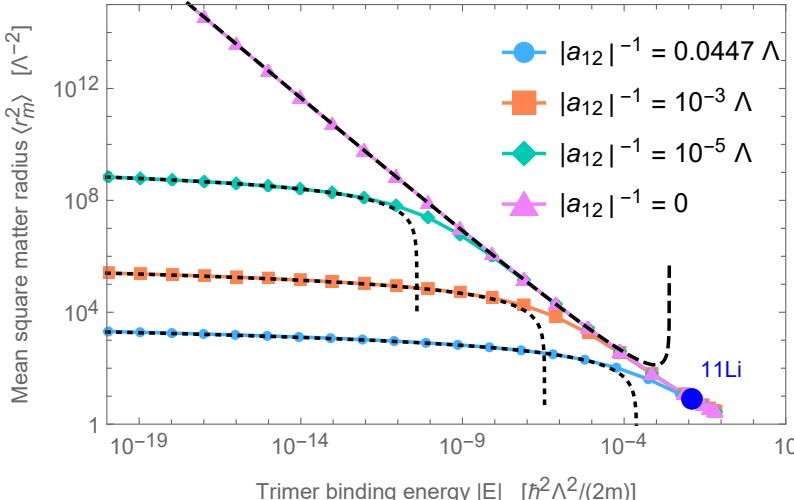

Figure 5: Mean square matter radius of the ground-state trimer as a function of its binding energy $|E|$, obtained numerically from the separable model. The different curves correspond to different values of the particle-particle scattering length $a_{12}$ shown by the vertical lines in Fig. 3. The dashed curve is obtained from the analytic formula Eq. (C.10) with $R_1$ set to $20\Lambda^{-1}$, and the dotted curves are obtained from the analytic formula Eq. (C.11) with $(R_1, R_2)$ set respectively to $(2.6, 16)10^4$, $(420, 1700)$, $(34, 64)$ in units of $\Lambda^{-1}$.

is repulsive. They are not related to the Efimov effect, i.e. the appearance of an attractive part (negative value of $s_n^2(R) = -|s_0|^2$) in the hyper-radial potential at shorter hyper-radius $R \ll |a_{12}|, |a_{23}|$. As a matter of fact, none of the analytical results presented here depend on the quantity $s_0$ characterising Efimov universality.

In addition, the above results apply to any Borromean halo state close to three-body dissociation. Thus, in a system where the Efimov effect occurs, they apply equally to the ground state and all Efimov states. In other words, the ground state and excited states have exactly the same geometry close to their dissociation threshold. This halo universality is explained by the fact that the hyper-angular part of the wave function is the same for all states close to their threshold, giving a universal shape distribution to all of these states. In contrast, the Efimov universality, which is the discrete scale invariance of the spectrum near the unitarity point, applies only approximately to the ground state because the hyper-radial part of its wave function significantly differs from the rescaled hyper-radial wave function of higher states, thus deviating from the discrete scaling invariance.

## 7 Observing halo universality

Two-neutron halo nuclei are prime candidates for the experimental evidence of halo universality, since neutrons are nearly resonant, and mean square radii can be extracted from experiments. Although the separable potential model is not a precise description of these halo nuclei, it can be used to estimate their universal nature. The case of the lithium-11 halo nucleus is represented as a blue point in Figs. 3, 4, and 5. This point is obtained by setting $a_{12}^{-1} = -0.0447\Lambda$ and $\Lambda^{-1} = 0.84$ fm to reproduce the neutron-neutron scattering length $a_{12} = -18.8$ fm and effective range $r_{12} = 2.83$ fm of the AV18 model [33], and setting $a_{23}^{-1} = -0.344|a_{23}^{(1)}|^{-1}$ to reproduce the two-neutron separation energy $|E| = 369.15(65)$ keV of lithium-11 [34]. These parameters lead to a mean-square radius $\langle R_{12,3}^2 \rangle = 4.70$ fm that is

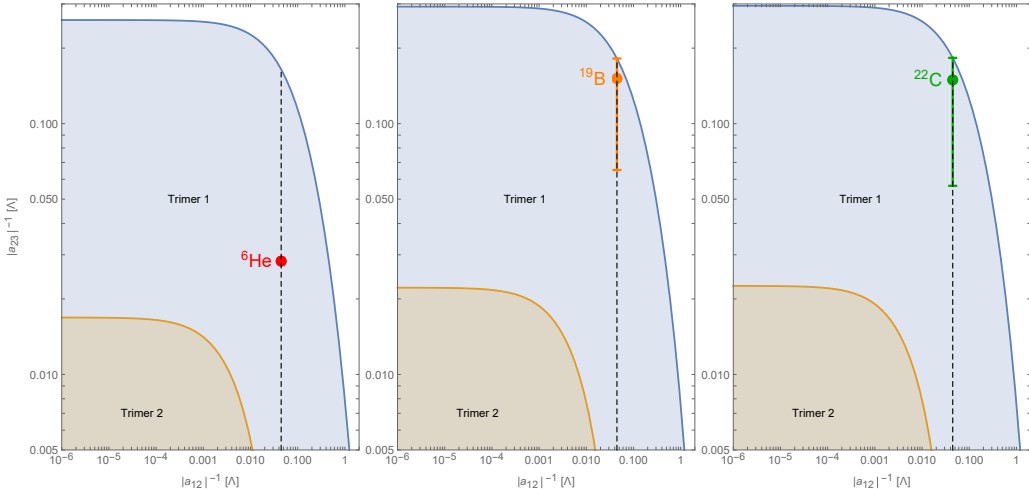

Figure 6: Same figure as Fig. 3 for mass ratio 4 (left), 17 (middle) and 20 (right). The points corresponding to helium-6, boron-19, and carbon-22 are obtained using the two-neutron separation energies $|E(^6\text{He})| = 975.45 \pm 0.05$ keV, $|E(^{19}\text{B})| = 90 \pm 560$ keV, and $|E(^{22}\text{C})| = 100 \pm 640$ keV from AME2016 [32].

in reasonable agreement with the value 5.01±0.32 fm [35] derived from experimental data. One can see from Fig. 4 that $^{11}$Li is not accurately described by the analytical limit Eq. (15), since its ratio $\langle r_m^2 \rangle / \langle r_c^2 \rangle$ is around 7.0 whereas Eq. (15) gives 6.3, i.e. an error of 10%. Also note that $^{11}$Li is yet too strongly bound to enter the regime of logarithmic divergence of its size, as shown in Fig. 5.

The situation of other two-neutron halo nuclei is illustrated in Fig. 6. In all cases, the core-neutron scattering length $a_{23}$ is set to reproduce the two-neutron separation energies compiled by AEM2016 [32]. The matter/core radius ratio $\langle r_m^2 \rangle / \langle r_c^2 \rangle$ of these halo nuclei is shown in Fig. 7. It appears that none of them lie in the regime where the analytical result of Eqs. (15, 21, 22) is accurate. A prominent reason is that the neutron-neutron scattering length $a_{12}$ is yet too small to fully reach this regime: it would need to be significantly larger to allow $a_{23}$ to approach the critical value $a_{23}^{(1)}$. Nevertheless, the cases of boron-19 and carbon-22 are the closest examples to halo universality, since their matter/core radius ratio is possibly less than a few percent off from the analytical formula. Further experimental determination of their binding energy and geometric properties could confirm this situation.

Ultracold mixtures of light and heavy atoms constitute another promising platform for the observation of halo universality. The advantage of such systems over atomic nuclei is that the scattering length between the light particles may be controlled by a magnetic Fano-Feshbach resonance. Examples include mixtures of caesium-133 and lithium-7 atoms, or lithium-6 atoms in different hyperfine states. However, experiments with these systems are known to be hindered by strong losses. In addition, specific experimental techniques such as Coulomb explosion imaging [36] should be implemented to measure geometrical properties such as the mean-square radii of ultracold atomic trimers.

# 8 Conclusion

In summary, this work presents a general picture of the geometric properties of trimer halos formed by a core and two particles, extending previous findings to the whole spectrum of three-body bound states, including Efimov states. It appears that close to the three-body dissociation

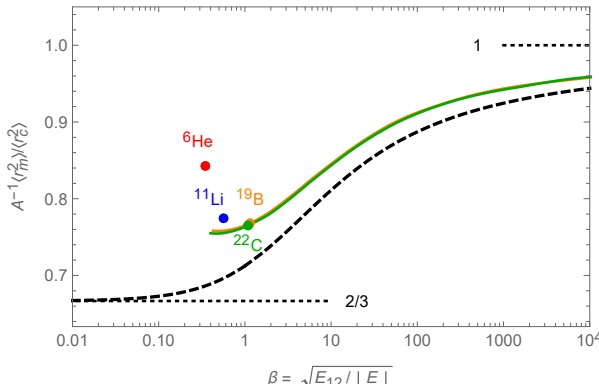

Figure 7: Matter/core radius ratio $\langle r_m^2 \rangle / \langle r_c^2 \rangle$ normalised by $A$ and energy ratio $\beta = \sqrt{E_{12}/|E|}$ for different halo nuclei. The solid curves indicate the possible values for boron-19 and carbon-22 due to the current uncertainty on their energy $E$, with $E_{12}$ being fixed by the neutron-neutron scattering length $a_{12} = -(0.0447\Lambda)^{-1}$ – this value is indicated by the vertical dashed lines in Fig. 6. The dashed curve corresponds to the analytical formula given by Eqs. (15, 21, 22).

threshold of any state, the trimer forms an extended halo with universal geometric properties that are given by the universal laws described in Ref. [11] away from the particle's resonance, and by the laws recently found in Ref. [24] close to the particles' resonance. Both limits can be understood in the hyper-spherical picture, which shows that the shape of the trimer is independent of its size at the three-body dissociation threshold. In the first (off-resonant) limit, the trimer's size increases logarithmically with vanishing trimer binding energy, while it increases almost as the inverse of the binding energy in the second (resonant) limit. This halo universality is independent of the Efimov effect, and thus applies indistinctly to all states, including Efimov states. It may be evidenced experimentally in two-neutron halo nuclei or ultracold atomic mixtures. Upon completion of this work, a related study [37] reported an extension of the formula Eq. (15) using effective-field theory techniques.

## Acknowledgments

He is thankful to L. Happ, M. Hongo, L. Pricoupenko, and J. Dalibard for helpful discussions.

**Funding information** The author acknowledges support from the JSPS Grants-in-Aid for Scientific Research on Innovative Areas (No. JP18H05407).

## A  Low-momentum and energy T-matrix element Eq. (13)

The Lippman-Schwinger Eq. (4) defining the two-body transition operator can be written explicitly (dropping here the indices $ij$),

$$\langle \boldsymbol{k} | \hat{T}(z) | \boldsymbol{q} \rangle = \langle \boldsymbol{k} | \hat{V} | \boldsymbol{q} \rangle + \int \frac{d^3 \boldsymbol{k}'}{(2\pi)^3} \langle \boldsymbol{k} | \hat{V} | \boldsymbol{k}' \rangle \frac{\langle \boldsymbol{k}' | T(z) | \boldsymbol{q} \rangle}{z^+ - \frac{\hbar^2 k'^2}{2\mu}} , \quad (A.1)$$

with $z^+ \equiv z + i\epsilon$. The interaction $\hat{V}$ (and thus $\hat{T}$) having a finite range $\Lambda^{-1}$, one can set $\boldsymbol{k} \approx \boldsymbol{0}$ and $\boldsymbol{q} \approx \boldsymbol{0}$ in the matrix elements for $k, q \ll \Lambda$ within an error of order $O(k^2, q^2)$. This gives

$$\frac{1}{\langle \mathbf{0}|\hat{T}(z)|\mathbf{0}\rangle} = \frac{1}{\langle \mathbf{0}|\hat{V}|\mathbf{0}\rangle} + \int \frac{d^3 \mathbf{k}}{(2\pi)^3} \Upsilon(\mathbf{k}, z) \frac{1}{\frac{\hbar^2 k^2}{2\mu} - z^+} , \tag{A.2}$$

with the function

$$\Upsilon(\mathbf{k}, z) \equiv \frac{\langle \mathbf{0}|\hat{V}|\mathbf{k}\rangle}{\langle \mathbf{0}|\hat{V}|\mathbf{0}\rangle} \frac{\langle \mathbf{k}|T(z)|\mathbf{0}\rangle}{\langle \mathbf{0}|\hat{T}(z)|\mathbf{0}\rangle} \xrightarrow[k \ll \Lambda]{} 1 . \tag{A.3}$$

Now, adding a counter-term in the integral, one can write:

$$\frac{1}{\langle \mathbf{0}|\hat{T}(z)|\mathbf{0}\rangle} = \frac{1}{\langle \mathbf{0}|\hat{V}|\mathbf{0}\rangle} + \int \frac{d^3 \mathbf{k}}{(2\pi)^3} \Upsilon(\mathbf{k}, z) \left( \frac{1}{\frac{\hbar^2 k^2}{2\mu} - z^+} - \frac{1}{\frac{\hbar^2 k^2}{2\mu}} \right) + \int \frac{d^3 \mathbf{k}}{(2\pi)^3} \Upsilon(\mathbf{k}, z) \frac{1}{\frac{\hbar^2 k^2}{2\mu}} . \tag{A.4}$$

Taking the limit $z \to 0$ gives

$$\frac{1}{\langle \mathbf{0}|\hat{T}(0)|\mathbf{0}\rangle} = \frac{1}{\langle \mathbf{0}|\hat{V}|\mathbf{0}\rangle} + \int \frac{d^3 \mathbf{k}}{(2\pi)^3} \Upsilon(\mathbf{k}, 0) \frac{1}{\frac{\hbar^2 k^2}{2\mu}} , \tag{A.5}$$

so that one can eliminate $\langle \mathbf{0}|\hat{V}|\mathbf{0}\rangle$ in favour of $\langle \mathbf{0}|\hat{T}(0)|\mathbf{0}\rangle$,

$$\frac{1}{\langle \mathbf{0}|\hat{T}(z)|\mathbf{0}\rangle} = \frac{1}{\langle \mathbf{0}|\hat{T}(0)|\mathbf{0}\rangle} + \int \frac{d^3 \mathbf{k}}{(2\pi)^3} \Upsilon(\mathbf{k}, z) \left( \frac{1}{\frac{\hbar^2 k^2}{2\mu} - z^+} - \frac{1}{\frac{\hbar^2 k^2}{2\mu}} \right)$$
$$+ \int \frac{d^3 \mathbf{k}}{(2\pi)^3} (\Upsilon(\mathbf{k}, z) - \Upsilon(\mathbf{k}, 0)) \frac{1}{\frac{\hbar^2 k^2}{2\mu}} . \tag{A.6}$$

Finally, one can consider the low-energy limit $z \ll \hbar^2 \Lambda^2 / 2\mu$. In this limit, by virtue of Eq. (A.3), the second line of Eq. (A.6) can be approximated as

$$\underbrace{\Upsilon(\mathbf{0}, z)}_{1} \frac{2\mu}{\hbar^2} \int \frac{d^3 \mathbf{k}}{(2\pi)^3} \left( \frac{1}{k^2 - \frac{2\mu z^+}{\hbar^2}} - \frac{1}{k^2} \right) = \frac{2\mu}{4\pi\hbar^2} i \sqrt{\frac{2\mu z}{\hbar^2}} ,$$

while the third line of Eq. (A.6) may be neglected, assuming that $\Upsilon(\mathbf{k}, z) - \Upsilon(\mathbf{k}, 0) \sim O(z)$. Using the standard result $\langle \mathbf{0}|\hat{T}(0)|\mathbf{0}\rangle = 4\pi\hbar^2 a / (2\mu)$ where $a$ is the scattering length, one obtains Eq. (13), which is valid with an error of order $O(k^2, k'^2, z_{ij})$.

# B  Functions $f_n$ anf $f_c$

## B.1  Numerator of Eq. (11)

Retaining only the Faddeev component $\mathcal{F}_{12} \equiv \mathcal{F}$ associated with the two particles in Eq. (2), the integral in the numerator of Eq. (11) is expressed as

$$\int d^3 K_{23,1} d^3 k_{23} \left| \nabla_{K_{23,1}} \frac{\mathcal{F}\left( |k_{23} - \frac{A}{A+1} K_{23,1}| \right)}{\frac{\hbar^2}{2\mu_{23,1}} K_{23,1}^2 + \frac{\hbar^2}{2\mu_{23}} k_{23}^2 - E} \right|^2 , \tag{B.1}$$

where we used $K_{12,3} = k_{23} - \frac{A}{A+1} K_{23,1}$. Using the relation

$$\nabla_p \mathcal{F}(|\alpha p + \beta q|) = \alpha \frac{\alpha p + \beta q}{|\alpha p + \beta q|} \mathcal{F}'(|\alpha p + \beta q|) , \tag{B.2}$$

and re-expressing the integrand in terms of $k \equiv k_{12}$ and $K \equiv K_{12,3}$ by using $K_{23,1} = -k_{12} - \frac{1}{2}K_{12,3}$, one arrives at

$$\int d^3K d^3k \left| -\left(\frac{A}{A+1}\right) \frac{\frac{K}{K}\mathcal{F}'(K)}{\frac{\hbar^2}{2\mu_{12,3}}K^2 + \frac{\hbar^2}{2\mu_{12}}k^2 - E} - \frac{\frac{\hbar^2}{2\mu_{23,1}}2\left(-k - \frac{1}{2}K\right)\mathcal{F}(K)}{\left(\frac{\hbar^2}{2\mu_{12,3}}K^2 + \frac{\hbar^2}{2\mu_{12}}k^2 - E\right)^2} \right|^2. \tag{B.3}$$

Expanding the square, the cross term proportional to $K \cdot k$ averages to zero since the orientations of $K$ and $k$ are independent. Integrating the remaining terms over $k$ by using

$$\int_0^\infty k^2 dk \frac{1}{(k^2 + Q^2)^2} = \frac{\pi}{4Q}, \tag{B.4}$$

$$\int_0^\infty k^2 dk \frac{1}{(k^2 + Q^2)^3} = \frac{\pi}{16Q^3}, \tag{B.5}$$

$$\int_0^\infty k^2 dk \frac{1}{(k^2 + Q^2)^4} = \frac{\pi}{32Q^5}, \tag{B.6}$$

$$\int_0^\infty k^2 dk \frac{k^2}{(k^2 + Q^2)^4} = \frac{\pi}{32Q^3}, \tag{B.7}$$

one arrives at

$$\boxed{\left(\pi \frac{2\mu_{12}}{\hbar^2}\right)^2 \sqrt{\frac{\mu_{12,3}}{\mu_{12}}} \left[\left(\frac{A}{A+1}\right)^2 f_c + \frac{A(A+2)}{A+1} f_n\right],} \tag{B.8}$$

with

$$f_c \equiv \int d^3K \left(\frac{\mathcal{F}'(K)^2}{\tilde{K}} - \frac{K\mathcal{F}'(K)\mathcal{F}(K)}{\tilde{K}^3} + \frac{K^2\mathcal{F}(K)^2}{2\tilde{K}^5}\right), \tag{B.9}$$

$$f_n \equiv \frac{1}{2}\int d^3K \frac{\mathcal{F}(K)^2}{\tilde{K}^3}, \tag{B.10}$$

and $\tilde{K}^2 \equiv K^2 - \frac{2\mu_{12,3}E}{\hbar^2}$.

## B.2  Numerator of Eq. (12)

Retaining only the Faddeev component $\mathcal{F}_{12} \equiv \mathcal{F}$ associated with the two particles in Eq. (2), the integral in the numerator of Eq. (12) is expressed as

$$\int d^3K d^3k \left| \nabla_K \frac{\mathcal{F}(K)}{\frac{\hbar^2}{2\mu_{12,3}}K^2 + \frac{\hbar^2}{2\mu_{12}}k^2 - E} \right|^2, \tag{B.11}$$

where $K = K_{12,3}$ and $k = k_{12}$. This gives

$$\int d^3K d^3k \left(\frac{\mathcal{F}'(K)}{\frac{\hbar^2}{2\mu_{12,3}}K^2 + \frac{\hbar^2}{2\mu_{12}}k^2 - E} - \frac{\frac{\hbar^2}{2\mu_{12,3}}2K\mathcal{F}(K)}{\left(\frac{\hbar^2}{2\mu_{12,3}}K^2 + \frac{\hbar^2}{2\mu_{12}}k^2 - E\right)^2}\right)^2. \tag{B.12}$$

Expanding the square and integrating over $k$, one arrives at:

$$\boxed{\left(\pi \frac{2\mu_{12}}{\hbar^2}\right)^2 \sqrt{\frac{\mu_{12,3}}{\mu_{12}}} f_c,} \tag{B.13}$$

where we used Eq. (B.4-B.6).

### B.3 Ratio of mean square radii

Using Eqs. (B.8) and (B.13) in the expression of the ratio of mean square radii Eq. (10), one obtains

$$\frac{\langle r_m^2 \rangle}{\langle r_c^2 \rangle} = \frac{\frac{1}{2}(A+1)^2 \frac{\langle R_{23,1}^2 \rangle}{\langle R_{12,3}^2 \rangle} + A}{A+2}$$
$$= \frac{1}{2}A\left(1 + \frac{f_n}{f_c}\right),$$

which yields Eq. (15) of the main text.

### B.4 Simplification of $f_n$ and $f_c$

The functions $f_n$ and $f_c$ given by Eqs. (B.9-B.10) can be expressed with the dimensionless variable $q \equiv \sqrt{\frac{\hbar^2}{2\mu_{12,3}|E|}}K$,

$$f_n(\beta) \propto \frac{1}{2} \int d^3q \frac{\mathcal{F}(q)^2}{\tilde{q}^3}, \tag{B.14}$$

$$f_c(\beta) \propto \int d^3q \left(\frac{\mathcal{F}'(q)^2}{\tilde{q}} - \frac{q\mathcal{F}'(q)\mathcal{F}(q)}{\tilde{q}^3} + \frac{q^2\mathcal{F}(q)^2}{2\tilde{q}^5}\right), \tag{B.15}$$

where $\tilde{q}^2 \equiv q^2 + 1$, and

$$\mathcal{F}(q) \propto \frac{1}{\beta + \tilde{q}}. \tag{B.16}$$

The function $f_c$ can be simplified as

$$f_n(\beta) \propto 2\pi \int_0^\infty \frac{q^2 dq}{\tilde{q}^3(\beta + \tilde{q})^2}, \tag{B.17}$$

$$f_c(\beta) \propto 2\pi \int_0^\infty \frac{dq}{\tilde{q}(\beta + \tilde{q})^2}. \tag{B.18}$$

The last expression can be verified by successive integrations by parts of Eq. (B.17), $\int u'v = -\int uv'$, with first $u(q) = q$, $v(q) = \tilde{q}^{-1}(\beta + \tilde{q})^{-2}$, and then $u(q) = q^3/3$; $v(q) = \tilde{q}^{-3}(\beta + \tilde{q})^{-2} + 2\tilde{q}^{-2}(\beta + \tilde{q})^{-3}$, leading to the original form of $f_c$ in Eq. (B.15). Finally, making the change of variable $y \equiv q^2 + 1$, one arrives at Eqs. (18-19) of the main text.

## C  Hyper-spherical representation

In the hyper-spherical representation, the values of the scattering lengths $a_{23} = a_{31}$ and $a_{12}$ for the corresponding pairs can be imposed in the wave function Eq. (23) by applying Bethe-Peierls conditions [29,30]. This leads to the following equations on the coefficients $\lambda^{(1)} = \lambda^{(2)}$ and $\lambda^{(3)}$:

$$M_{11}\lambda^{(1)} + M_{13}\lambda^{(3)} = 0, \tag{C.1}$$
$$M_{31}\lambda^{(1)} + M_{33}\lambda^{(3)} = 0,$$

with

$$M_{11} = \left(-s\cos\left(s\frac{\pi}{2}\right) + \frac{R}{a_{23}}\sin\left(s\frac{\pi}{2}\right)\right) + 2\frac{\sin s\gamma}{\sin 2\gamma}, \tag{C.2}$$

$$M_{13} = 2\frac{\sin s\gamma'}{\sin 2\gamma'}, \tag{C.3}$$

$$M_{31} = 4\frac{\sin s\gamma'}{\sin 2\gamma'}, \tag{C.4}$$

$$M_{33} = -s\cos\left(s\frac{\pi}{2}\right) + \frac{R}{a_{12}}\sin\left(s\frac{\pi}{2}\right). \tag{C.5}$$

To obtain non-zero solutions $\lambda^{(1)}$ and $\lambda^{(3)}$, the determinant of Eq. (C.1) should be zero, which leads to

$$\left(-\cos\left(s\frac{\pi}{2}\right) + \frac{2}{s}\frac{\sin(s\gamma)}{\sin 2\gamma} + \frac{\sin\left(s\frac{\pi}{2}\right)}{s}\frac{R}{a_{23}}\right)\left(-\cos\left(s\frac{\pi}{2}\right) + \frac{\sin\left(s\frac{\pi}{2}\right)}{s}\frac{R}{a_{12}}\right) = 2\left(\frac{2\sin(s\gamma')}{s\sin 2\gamma'}\right)^2, \tag{C.6}$$

which is Eq. (26) of the main text.

## C.1 Mean-square radii for vanishing binding energy

In the hyper-spherical representation, any mean square radius is given by an expression of the form,

$$\langle r^2 \rangle = \frac{\sum_{n,n'}\cdots\int dR\, R^2 \left(\sqrt{R}F_n(R)\right)\left(\sqrt{R}F_{n'}(R)\right)}{\sum_{n,n'}\cdots\int dR \left(\sqrt{R}F_n(R)\right)\left(\sqrt{R}F_{n'}(R)\right)}, \tag{C.7}$$

where the dots indicate numerical factors resulting from integration over the hyper-angles. From Eq. (24) of the main text, at sufficiently large hyper-radius $R \gg R_0$ (to be specified below), the hyper-radial functions $F_n(R)$ satisfy the following equation:

$$\left(-\partial_R^2 + \frac{s_n^2(\infty) - 1/4}{R^2} - \frac{2mE}{\hbar^2}\right)\sqrt{R}F_n(R) = 0, \tag{C.8}$$

where $s_n(\infty) = \lim_{R\to\infty} s_n(R)$. It follows that

$$F_n(R) \xrightarrow[R\gg R_0]{} K_{s_n(\infty)}(\kappa R), \tag{C.9}$$

where $\kappa = -2mE/\hbar^2$, and $K$ designates the modified Bessel function of the second kind.

**Off-resonance case $1/|a_{12}| \neq 0$**

The solutions $s_n$ of Eq. (C.6) behave as $s_n(R) \xrightarrow[R\gg R_0]{} 2n$, with $R_0 \sim \max(|a_{12}|, |a_{23}|)$. In this case, only the lowest channel $n = 1$ gives a hyper-radial function that extends far beyond $R_0$ for vanishing energy $E$. Close to the three-body dissociation threshold, one can thus neglect the other channels in the calculation of the mean square radius Eq. (C.7), and using Eq. (C.9) one finds

$$\langle r^2 \rangle \xrightarrow[\kappa\ll R_0^{-1}]{} \eta\langle R^2\rangle = \eta R_1^2 \ln\frac{1}{\kappa R_2}, \tag{C.10}$$

where the distances $R_1, R_2 \gtrsim R_0$ are independent of $\kappa$. This shows that the mean square radius diverges logarithmically with vanishing trimer energy. The coefficient $\eta$ results from the hyper-angular integration.

**On-resonance case $1/|a_{12}| = 0$**

The solutions $s_n$ of Eq. (C.6) behave as $s_n(R) \xrightarrow[R \gg R_0]{} n$, with $R_0 \sim |a_{23}|$. In this case, both the lowest channel $n = 1$ and second channel $n = 2$ give hyper-radial functions that extend far beyond $R_0$ with vanishing energy. However, the channel $n = 1$ extends to distances $\sim \kappa^{-1}$, whereas the $n = 2$ channel extends to distances $\sim R_0 \ln \frac{1}{\kappa R_0}$. Close to the three-body dissociation threshold, the mean square radius is therefore dominated again by the lowest channel $n = 1$, which gives

$$\langle r^2 \rangle \xrightarrow[\kappa \ll R_0^{-1}]{} \eta \langle R^2 \rangle = \eta \frac{2}{3\kappa^2} \left( \ln \frac{1}{\kappa R_1} \right)^{-1}, \tag{C.11}$$

where the distance $R_1 \gtrsim R_0$ is independent of $\kappa$. This shows that the mean square radius diverges slightly more slowly than the inverse of the vanishing energy $E$.

## C.2  Calculation of the matter and the core radii

Let us now calculate the matter and core radius explicitly. The coefficients $\eta$ for the matter and core radii follow from Eq (6-9) of the main text:

$$\eta_m = 2 \frac{(A+1) \langle \cos^2 \alpha_1 \rangle + \langle \cos^2 \alpha_3 \rangle}{(A+2)^2}, \tag{C.12}$$

$$\eta_c = \frac{2}{A(A+2)} \langle \cos^2 \alpha_3 \rangle. \tag{C.13}$$

**Off-resonance case $1/|a_{12}| \neq 0$**

In the dominant hyper-angular channel, $s_1(R) \to 2$, therefore the corresponding component is independent of hyper-angles, such that $\langle \cos^2 \alpha_i \rangle = 1/2$. Therefore,

$$\eta_m = \frac{1}{A+2}, \tag{C.14}$$

$$\eta_c = \frac{1}{A(A+2)}. \tag{C.15}$$

It follows that

$$\frac{\langle r_m^2 \rangle}{\langle r_c^2 \rangle} = \frac{\eta_m \langle R^2 \rangle}{\eta_c \langle R^2 \rangle} = A. \tag{C.16}$$

**On-resonance case $1/|a_{12}| = 0$**

From Eq. (C.1), one has

$$\frac{\lambda^{(3)}}{\lambda^{(1)}} = -\frac{M_{11}}{M_{13}}, \tag{C.17}$$

and in the limit $R \to \infty$, for the dominant hyper-angular channel $s_1(R) \to 1$, one finds

$$\frac{\lambda^{(3)}}{\lambda^{(1)}} = -\frac{\frac{R}{a_{23}} + \frac{1}{\cos \gamma}}{\frac{1}{\cos \gamma'}} \to \infty. \tag{C.18}$$

Thus, in the calculation of hyper-angular averages, one can only retain the component

$$\Phi^{(3)}(\alpha; R) \propto \frac{1}{\sin \alpha} \tag{C.19}$$

(It is the same approximation as retaining only the Faddeev component $\mathcal{F}_{12}$ in the calculation of Sec. 4). This gives,

$$\langle \cos^2 \alpha_3 \rangle = \frac{\int_0^{\pi/2} d\alpha_3 \sin^2(2\alpha_3) \int_{-1}^1 du_3 \left( \cos^2 \alpha_3 \right) \left| \Phi^{(3)}(\alpha_3) \right|^2}{\int_0^{\pi/2} d\alpha_3 \sin^2(2\alpha_3) \int_{-1}^1 du_3 \left| \Phi^{(3)}(\alpha_3) \right|^2}$$
$$= \frac{\int_0^{\pi/2} d\alpha \cos^4 \alpha}{\int_0^{\pi/2} d\alpha \cos^2 \alpha} = \frac{3}{4}, \tag{C.20}$$

and

$$\langle \cos^2 \alpha_1 \rangle = \frac{\int_0^{\pi/2} d\alpha_3 \sin^2(2\alpha_3) \int_{-1}^1 du_3 \left( \cos^2 \alpha_1 \right) \left| \Phi^{(3)}(\alpha_3) \right|^2}{\int_0^{\pi/2} d\alpha_3 \sin^2(2\alpha_3) \int_{-1}^1 du_3 \left| \Phi^{(3)}(\alpha_3) \right|^2}$$
$$= \frac{1}{4} \left( \frac{1 + 2A}{1 + A} \right), \tag{C.21}$$

where we used the change of variables $\int_{-1}^1 du_3 = \frac{2}{\sin 2\gamma'} \int_{|\frac{\pi}{2} - \alpha_3 - \gamma'|}^{\frac{\pi}{2} - |\alpha_3 - \gamma'|} \frac{\sin 2\alpha_1}{\sin 2\alpha_3} d\alpha_1$ and the definitions of $\gamma'$ and $\gamma$ in terms of $A$. From Eqs. (C.12-C.13) one finds

$$\eta_m = \frac{1}{A + 2}, \tag{C.22}$$

$$\eta_c = \frac{3}{2} \frac{1}{A(A + 2)}. \tag{C.23}$$

It follows that

$$\boxed{\frac{\langle r_m^2 \rangle}{\langle r_c^2 \rangle} = \frac{\eta_m \langle R^2 \rangle}{\eta_c \langle R^2 \rangle} = \frac{2}{3} A.} \tag{C.24}$$

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
