# Peer review of "Universal geometry of two-neutron halos and Borromean Efimov states close to dissociation"

_SciPost Physics, doi:SciPost Phys. 15, 123 (2023)_

## Round 1 · Referee Report · Anonymous (Referee 1) · 2023-5-13

Strengths

See attached report

Weaknesses

See attached report

Report

See attached file

Requested changes

See attached file

Attachment

  • validity: high
  • significance: high
  • originality: high
  • clarity: ok
  • formatting: good
  • grammar: good

Author:  Pascal Naidon  on 2023-07-11  [id 3794]

(in reply to Report 1 on 2023-05-13)

I would like to thank the referee for their very thorough review.

(1) The author refers in the title to Borromean Efimov states. To my understanding Efimov states are always Borromean, so this statement seems to be redundant. Is there some particular distinction for Borromean Efimov states?

In fact, Efimov states are not always Borromean. When one of the interactions supports a weakly bound two-body state, implying a positive scattering length for this interaction, the Efimov states are not Borromean, by definition of the term ‘Borromean’ as ‘bound in the absence of bound state for any subsystem’. The present study is restricted to negative scattering lengths, for which Efimov states are Borromean. It is therefore important to specify this point in the title.

(2) To my understanding, the analytical relations found in the work based on effective field theory [Ref. 24 in the manuscript] apply only to the ground trimer state. If that is the case, I would suggest to include this statement at the last sentence of the first paragraph of the Introduction.

That is a very good point. The work based on effective theory may indeed appear to be intended to describe a ground state, although it does apply to excited states as well. This is precisely a point clarified in this manuscript. I have included a sentence in the second paragraph to emphasise this point.

(3) It would help the reader if the range of the interactions Lambda was introduced in Equation (1), instead of introducing it later in the text [Section 4].

Following the referee’s suggestion, the range is now introduced after equation (1).

(4) Regarding the three pairwise interactions, the author assumes later in the text (beginning of chapter 5) that Lambda is the same both for neutron-neutron and neutron-core interactions. Are the two interactions of the same form, or is there some reason why the range of interactions can be treated to be the same?

The referee is correct that the ranges of these interactions should be different in general. However, the interactions being of the same physical nature (nuclear force for nucleons, atomic forces for atoms), their ranges must be of the same order of magnitude. They are therefore taken to be equal in the numerical calculations for simplicity. This is now explained in the manuscript. As a check of the validity of this assumption, the resulting model is applied to the case of lithium-11, which yields mean-square radii in agreement with experimental data.

(5) Right before Eq. (10), where the Jacobi vectors Rij,k are introduced, it would be good if there was a reference to Fig. 1.

The reference to Fig. 1 was added.

(6) How is the low-energy expression for the T-matrix elements derived? If the steps have been already carried out in a different publication, I suggest to cite these works.

Although this expression looks like the textbook result for the on-shell T-matrix elements, it is in fact the general off-shell expression necessitated by the three-body problem. I could not find a reference deriving this expression explicitly, therefore I added the general derivation in an appendix.

(7) I think there is a typo in the inline formula after Eq. (13). I think it should read,

$$\sqrt{2 \mu_{12} z_{12}/ \hbar^2} << a^{-1}{12}<< a^{-1}.$$},a^{-1}_{31

I thank the referee for spotting the typo. In fact, this condition was not correct, as pointed out by the second referee, and it has been corrected.

(8) The author argues before Eq. (14) that the Faddeev component associated to the neutron-neutron dimer is the dominant one. Is that the case due to its dependence on the inverse square root of the two-body neutron-neutron energy, which is assumed to be very small?

Yes, it is precisely for this reason.

(9) Do the results in Section 4 apply only to the ground state?

The derivation of Section 4 does not require the bound state to be a ground state, so it applies to any excited bound state as well. This is now stated explicitly at the beginning of Section 4.

(10) In Section 4, the author distinguishes between two limiting cases for the matter over core mean-square radii, providing results depending solely on A, the mass ratio. How do these limits affect the geometry of the three-body system, and what are the imposed relations on the Jacobi vectors Rij,k ?

It is quite straightforward from Eq. (10) to find the constraint on Rij,k for a given limit of the ratio <r_m^2>/<r_c^2>. More generally, any geometric property in these limits will have some analytic expression depending on A, although it may look a bit more complicated. The choice of r_m and r_c just gives one of the simplest expressions.

(11) Why does the author consider only the case A=10 as a mass ratio between the core and the neutrons? Does it refer to a particular system, or is it a prototype system, and larger or smaller mass ratios lead to the same phenomenology?

Indeed, the choice A=10 was purely arbitrary. In the revised manuscript, the case A=9 is now considered, which can be applied to the halo nucleus of lithium-11 and thus gives an illustration with a physical example. Other halo nuclei with different mass ratios are also discussed in the revised manuscript.

(12) In Fig. 2, at any fixed inverse neutron-neutron scattering length, an infinity of trimer states appears due to the Efimov effect, as the neutron-core length is tuned to larger values. Why only five states appear in the leftmost corner of both panels? Is it due to numerical difficulties as the energy of the states becomes smaller and smaller, or does the scaling factor become very large?

It is true that there is an infinity of trimer states, but only five of them appear in the range of the plots shown in Fig. 2. The other states, which would lie outside of these plots, are thus not missing (although the region of the sixth state is admittedly very close to the boundary of the plot).

(13) Is it true that for highly excited trimers [upper panel and leftmost part of Fig. 2], the scaling factor for the dissociation thresholds becomes the same as for three identical particles?

Actually, it is not the same scaling factor as for three identical particles (~22.7) because the mass ratio is different, although it is very close (~17.6). The small variation of the scaling factor with mass ratio can be seen in the lowest curve in Fig. 6.2 of arxiv/1610.09805v3.

(14) In the caption of Fig. 3, the last sentence should refer to $|a^{-1}_{12}|$ and not $|a^{-1}_{23}|$ ?

I thank the referee for spotting this typo. It has now been corrected.

(15) In Fig. 4, the good agreement with the analytic formula for beta go to zero, applies only for neutron-core scattering lengths such that the system is close to the three-body dissociation threshold. The author states “Therefore, it appears that the analytical formula does require a fine tuning of the core-particle interaction”. In PRL 128, 212501 (2022) however, it seems that the formula applies only close to the three-body dissociation threshold and neutron-neutron two-body resonance. Where does the fine tuning come from? It appears that in the current work, the author tests the range of applicability of the formula derived in PRL 128, 212501 (2022), providing bounds for its validity.

As shown in Fig. 4, the formula works well when the neutron-core scattering length $a_{23}$ is within 1% of the values $a_{23}^{(i)}$ but becomes inaccurate for other values of $a_{23}$. In this sense, the formula requires a rather fine-tuning of the core-particle interaction. However, the refereee is correct that this could be interpreted as the mere requirement of the binding energy being small enough. Therefore, this sentence on the fine tuning of $a_{23}$ may be removed in the final version if the referee deems it confusing.

(16) It would be less confusing if the horizontal axis in Fig. 4, was just the neutron-neutron scattering length. My understanding is that both the a12 scattering length as well as the binding energy |E| vary as one follows the horizontal dashed lines sketched in Fig. 3. Is therefore the matter-to-core radii depicted with respect to beta out of convenience? So that a comparison with the analytical formulas is straightforward?

Indeed, varying beta corresponds to following the horizontal dashed lines of Fig. 3. The reason for using $\beta$ instead of $a_{12}$ is that it is indeed the variable appearing in the analytical formula, and it also allows a direct comparison between different states. The halo universality close to the threshold would be less apparent if the variable $a_{12}$ was used.

(17) Do the curves for |a23| away from the three-body threshold come closer to the 2A/3 value in Fig. 4, when one considers smaller values of beta (10^-7) ? Or do they already saturate as suggested by the presented values?

As far as it is possible to tell from the numerical calculations, the curves do not come close to the value 2A/3 and indeed saturate as suggested by the figure.

(18) How does the choice of separable interactions or the Gaussian form factors affect the limit of beta go to zero in the numerical calculations presented in Fig. 4? Are there small deviations if one chooses other form factors?

That is a very good question. I have not checked what happens for other form factors or non-separable interactions, as I was primarily interested in checking the analytical limit in this work. It is possible that there is further model-independence beyond the analytically computable results.

(19) Regarding Eqs. (23) and (24) for the hyper-spherical formalism, I suggest to provide a few references regarding the method.

It is now specified that the formalism is called the "hyper-spherical adiabatic expansion" and a comprehensive reference is given.

(20) I suggest to move Section 6 at the beginning, after the introduction of the model. In that regard, a neat explanation is provided for the universal relations of the radii for the ground state, along with an extension to excited states. I think it would be better to present these results directly, to make the distinction with PRL 128, 212501 (2022) more clear.

I understand the referee’s logic to have all the analytical results presented first, and finally illustrate them with numerical calculations. However, I am afraid that Section 6 would appear too abstract without first presenting the general picture obtained with the numerical calculations. I feel it is better to keep Section 6 as the section presenting a general interpretation of both the analytical formula and the numerical calculations.

(21) The results obtained in Section 6 apply close to the zero energy three-body threshold? If yes, it would be helpful for the reader to state that explicitly.

The beginning of the last paragraph has been rephrased to make this point explicit.

(22) The sentence at the end of Section 6 is a bit confusing. From the results obtained within the hyper-spherical formalism, it is shown that the limiting cases of the mean-square radii apply to all states. To which lack of Efimov universality does the author refer to?

The Efimov universality is a discrete invariance of the spectrum near unitarity. The ground state trimer conforms only approximately to this invariance. If one picks a highly-excited state, it can be scaled very precisely onto another highly-excited state, but it would not superimpose very well when scaled onto the ground state. In contrast, the universality presented in this work is the same for the ground state and excited state, as it concerns only the shape, and not the hyper-radial distribution. I have rephrased the explanation more clearly, and now refer to this universality as “halo universality” to make the distinction with “Efimov universality” more explicit.

---

## Round 1 · Referee Report · Anonymous (Referee 2) · 2023-5-22

Strengths

The author investigates the geometry of three-body halos consisting of a
a core and two neutrons for different values of the neutron-core
interaction. He extends previous findings for systems with one
three-body state to systems with excited states. The universal relations
for the radii are first derived analytically using the Faddeev equations
and then verified in a numerical investigation with separable potentials.
In particular the analytical derivation is very instructive and transparent
and it is confirmed by the numerical investigation.
The paper is timely, interesting, and well written.

Weaknesses

I have a few comments that should be addressed before the paper can be published (see report).

Report

The author investigates the geometry of three-body halos consisting of a a core and two neutrons for different values of the neutron-core interaction. He extends previous findings for systems with one three-body state to systems with excited states. The universal relations for the radii are first derived analytically using the Faddeev equations and then verified in a numerical investigation with separable potentials. In particular the analytical derivation is very instructive and transparent and it is confirmed by the numerical investigation. The paper is timely, interesting, and well written. However, I have a few comments that should be addressed before the paper can be published:

1) The relations in Sec. 4 are derived under the assumption |1/a_12| << |1/a_13|, |1/a_23|. Thus I can see that the prediction for beta >> 1 is universal. But I would think that the prediction for beta << 1 does depend on the relative sizes of E and the energy scales determined by |1/a_13|, |1/a_23|, and is not valid unless sqrt(|E|) << |1/a_13|, |1/a_23 (which is the case considered in [24]). Only in this case I, would consider the limit beta << 1 to be universal.

2) Is the extension to excited states not implicitly contained in [24]? The EFT of [24] does not use in any way that the three-body halo is in its true ground state nor is this requirement stated anywhere. The presence of more deeply bound three-body states would not change the calculations in [24] in any way. Thus I find the results of the present study not that surprising. Am I missing something here?

3) I do not quite understand the claim that the universality is independent of the Efimov effect. This might very well be true, but I do not see how the numerical investigations show this. All states investigated in the numerical investigation are bound via the Efimov effect. In particular, the three-body states only exist above a critical value of the neutron-core scattering length. This is exactly the behavior of Efimov states. It is true that the true ground states of the separable model theory may have large corrections because their energies are close to the cutoff scale. Nevertheless, their behavior for small deviations of the pair scattering length (i.e. in the universal window) is that of Efimov states.

Requested changes

Address comments 1)-3) in report.

  • validity: high
  • significance: high
  • originality: good
  • clarity: high
  • formatting: excellent
  • grammar: excellent

Author:  Pascal Naidon  on 2023-07-11  [id 3795]

(in reply to Report 2 on 2023-05-22)

1) The relations in Sec. 4 are derived under the assumption |1/a_12| << |1/a_13|, |1/a_23|. Thus I can see that the prediction for beta >> 1 is universal. But I would think that the prediction for beta << 1 does depend on the relative sizes of E and the energy scales determined by |1/a_13|, |1/a_23|, and is not valid unless sqrt(|E|) << |1/a_13|, |1/a_23 (which is the case considered in [24]). Only in this case I, would consider the limit beta << 1 to be universal.

I thank the referee for bringing up this point. The assumption for Sec. 4 was indeed incorrectly stated. The proper assumption now reads $\sqrt{2\mu_{12}|E|}/\hbar , |1/a_{12}| << |1/a_{13}|, |1/a_{23}|$. This condition implies that $\sqrt{2\mu_{12}|E|}/\hbar << |1/a_{13}|, |1/a_{23}|$, as correctly stated by the referee, and that the Faddeev component $F_{12}$ is always dominant over the two other components at low momentum, which is the basis for the derivation of Sec 4. Note that this holds for any value of $\beta$, including $\beta \ll 1$ and $\beta \gg 1$. I hope this clarifies the conditions of validity.

2) Is the extension to excited states not implicitly contained in [24]? The EFT of [24] does not use in any way that the three-body halo is in its true ground state nor is this requirement stated anywhere. The presence of more deeply bound three-body states would not change the calculations in [24] in any way. Thus I find the results of the present study not that surprising. Am I missing something here?

Indeed, the results in that reference are implicitly applicable to the excited states, but there is no mention of excited nor Efimov states in that reference. It appears that the work has been understood by many to apply only to a ground state, as exemplified by the first referee's statement: "To my understanding, the analytical relations found in the work based on effective field theory [Ref. 24 in the manuscript] apply only to the ground trimer state". So it seems needed to emphasise the applicability of their formulas to excited states, and Efimov states in particular. To acknowledge that this point was implicitly contained in the original work, I added the sentence: "Although it is implicit in the work of Ref. [24], it is here emphasised that the analytical relations apply not only to a ground state but also to excited Borromean halo states".

3) I do not quite understand the claim that the universality is independent of the Efimov effect. This might very well be true, but I do not see how the numerical investigations show this. All states investigated in the numerical investigation are bound via the Efimov effect. In particular, the three-body states only exist above a critical value of the neutron-core scattering length. This is exactly the behavior of Efimov states. It is true that the true ground states of the separable model theory may have large corrections because their energies are close to the cutoff scale. Nevertheless, their behavior for small deviations of the pair scattering length (i.e. in the universal window) is that of Efimov states.

This is indeed a subtle point. It is true that the numerical investigation exhibits states which are all bound by the Efimov attraction. The main purpose of this numerical calculation is to show that the "halo universality" does apply to any Efimov state. On the other hand, their "halo universality" close their point of disssociation is governed by a repulsion at distances beyond the scattering length, i.e. beyond the range of the Efimov attraction. This repulsion always occurs, irrespective of the presence or absence of the Efimov effect. It is therefore independent of the Efimov effect. This is confirmed by the fact that none of the analytical derivations rely on the presence or absence of the Efimov effect, and do not depend on the quantity $s_0$ characterising Efimov universality. I expanded the discussion in the manuscript to make this important point clearer.

---

## Round 1 · Referee Report · Anonymous (Referee 3) · 2023-6-20

Strengths

Universality is a crucial ingredient in our attempt to patch together seemingly disparate effects in physics and create a unifying approach to understanding of these effects. In this context and in Efimov physics, universality refers to the ability of the effect to be insensitive to physics at short scales. This has led to celebrated universal scaling of Efimov exponents and energies and its discovery in atomic ultracold systems.

In this work, the author hints at another universal behavior, albeit this time a geometrical behavior with ratio of the masses with the core particle and the ratio of three-body (Efimov) energy and two-body (neutron-neutron) scattering length. This is a fairly complicated universal behavior whose utility remains to be understood or its practical applicability needs to be explored.

Weaknesses

The alluring beauty of Efimov universality lies in the concept of scattering length; a short-range collisional parameter, which determines the exponent of Efimov scaling and its all other correlated few-body behavior. No other parameters are necessary.

This work, while scientifically justified and interesting, applies universality to the length scales of interactions- inter-particle distances- and is informed by a ratio of radii in Eq. (15). Past work on mass ratios revealed certain magic ratios for which the original Efimov universality held.

The current universality is a novel concept, but how practically it can visualized and interpreted is not clear. Can such universalities be observed?

That the separation between hyper-radius (R) and hyper-angles (alpha) holds should not be a surprise.

The author refers to the applicability of this class of universality to "all states, including Efimov states". What does this mean? Do other molecular three-body states follow such universality?

What is a Borromean Efimov?

Report

This work is scientifically sound and should be considered for publication. It is not clear to this reviewer how such universality can be tested.
  • validity: good
  • significance: good
  • originality: high
  • clarity: good
  • formatting: good
  • grammar: good

Author:  Pascal Naidon  on 2023-07-11  [id 3796]

(in reply to Report 3 on 2023-06-20)

I thank the referee for pointing out the strenghs of the work. Regarding their concerns, here are my replies.

1) The alluring beauty of Efimov universality lies in the concept of scattering length; a short-range collisional parameter, which determines the exponent of Efimov scaling and its all other correlated few-body behavior. No other parameters are necessary. This work, while scientifically justified and interesting, applies universality to the length scales of interactions- inter-particle distances- and is informed by a ratio of radii in Eq. (15). Past work on mass ratios revealed certain magic ratios for which the original Efimov universality held.

As shown in the present work (and the previous work of Hongo and Son) the halo universality is also governed by a single parameter, the ratio $\beta$ between the three-body energy and the two-body energy.

It is important to note that the inter-particle distances in 3-body halo states (states close to three-body dissociation) is much larger than the length scale of interaction. This is why such sates exhibit universal properties. Efimov states (states close to two-body dissociation) are universal for the same reason: since the scattering length approaches infinity, Efimov states are much larger than the length scale of interaction and thus universal.

2) The current universality is a novel concept, but how practically it can visualized and interpreted is not clear. Can such universalities be observed?

The observability is indeed an important point. To observe halo universality, one should be able to probe the geometric properties of a three-body system near three-body dissociation. A new section discussing the observation of halo universality in halo nuclei and ultracold atoms has been added to the manuscript. The advantage of halo nuclei is that their geometry can be probed experimentally, but their disadvantage is that they only approach the regime of halo universality. The advantage of ultracold atomic trimers is that they can be tuned to fully reach the regime of halo universality, but determining their geometry experimentally is more difficult.

3) That the separation between hyper-radius (R) and hyper-angles (alpha) holds should not be a surprise.

In general, these two variables are not expected to be separable. There are only two cases where they become separable: - at the unitarity limit, i.e. at the two-body dissociation point, where the Efimov effect occurs. - near the three-body dissociation point, which is pointed out in the present work.

4) The author refers to the applicability of this class of universality to "all states, including Efimov states". What does this mean? Do other molecular three-body states follow such universality?

Yes, it applies to any three-body bound state as long as it is sufficiently close to three-body dissociation. The fact that it also applies to triatomic molecules close to dissociation has been emphasised in the abstract.

5) What is a Borromean Efimov?

A Borromean Efimov state is an Efimov state that is Borromean, i.e. it is an Efimov three-body bound state in the absence of two-body bound state. The Borromean regime for Efimov states is the regime of negative scattering lengths, for which there is no two-body bound state. This is the regime studied in this work.

---

## Round 2 · Referee Report · Anonymous (Referee 2) · 2023-7-19

Report

All comments from the first report have been addressed in a satisfactory fashion. I recommend the paper for publication in SciPost Physics.

---

## Round 2 · Referee Report · Anonymous (Referee 1) · 2023-7-21

# *Universal geometry of two-neutron halos and Borromean Efimov states close to dissociation*

I deem that the answers provided by the author are explanatory and address all of the raised questions. There is now a clearer distinction with the previously published work [PRL 128, 212501 (2022)] in terms of the extension of the results from the ground to any excited halo state. The presentation of the results is now more illuminating, and the concept of halo universality is now explained with great clarity. In that regard, I endorse publication of the manuscript as an article in the journal. There is however a remaining question, originating from one of the author's answer, which I would like to be addressed more out of curiosity,

1. In the answer of point (13), the author stated that the scaling ratio of the dissociation thresholds presented in the upper panel of Fig. 2 (leftmost part) is not the same as for three identical particles, albeit very close (~17.6), due to the mass difference. To my understanding the mass ratio in the upper panel of Fig. 2 is 1 (equal mass), and all of the interactions are nearly resonant. Where does the different scaling factor from three identical particles come from ? Is it due to the differences in the neutron-neutron and neutron-core scattering lengths ?

---

## Editorial Decision

published